# Generation of Multi-modal Brain Tumor MRIs with Disentangled Latent Diffusion Model

**Yoonho Na**[1]                                                    YOONHO94.NA@SNU.AC.KR
**Kyuri Kim**[1]                                                      KYURIKIM@SNU.AC.KR
**Sung-Joon Ye**[1]                                                   SYE@SNU.AC.KR
**Hwiyoung Kim**[2]                                                   HYKIM82@YUHS.AC
**Jimin Lee**[3]                                                      JIMINLEE@UNIST.AC.KR

[1] *Department of Applied Bioengineering, Graduate School of Convergence Science and Technology, Seoul National University, Seoul, Republic of Korea*

[2] *Department of Biomedical Systems Informatics and Center for Clinical Imaging Data Science, Yonsei University College of Medicine, Seoul, Republic of Korea*

[3] *Department of Nuclear Engineering, Ulsan National Institute of Science and Technology, Ulsan, Republic of Korea*

**Editors:** Under Review for MIDL 2023

## Abstract

Deep-learning based image generation methods have been widely used to overcome data deficiency. The same is true also as in medical field, where data shortage problem is frequent. In this study, we propose multi-modal brain tumor Magnetic Resonance Imaging (MRI) generation framework, called Disentangled Latent Diffusion Model (DLDM) to tackle data deficiency in medical imaging. We train an autoencoder that disentangles the feature of multi-modal MR images into modality-sharing and modality-specific representations. By utilizing the feature disentanglement learned from the autoencoder, we were able to train a diffusion model that can generate modality-sharing and modality-specific latent vector. We evaluate our approach with clean-FID and improved precision & recall. The results were compared with GAN-based model, StyleGAN2.

**Keywords:** Generation, Multi-modal, MRI, Feature disentanglement, Diffusion model.

## 1. Introduction

In this work, we propose a novel approach for generating multi-modal brain tumor MRIs using Diffusion Model(DM) with feature disentanglement. Existing methods for generating multi-modal MRIs typically rely on image-to-image translation and thus require a source image to obtain structural information. Our proposed model, which we call disentangled latent diffusion model (DLDM), is capable of generating modality-sharing and modality-specific information separately, eliminating the need for a source image. Using this approach, DLDM can generate unlimited number of multi-modal MR images by learned distribution of brain structures, in contrast to image-to-image translation based models that are limited to the brain structures present in the acquired data. To the best of our knowledge, no prior studies have utilized DMs for generating multi-modal MRIs and also, have generated multi-modal MRIs with fixed structure without any use of source image.

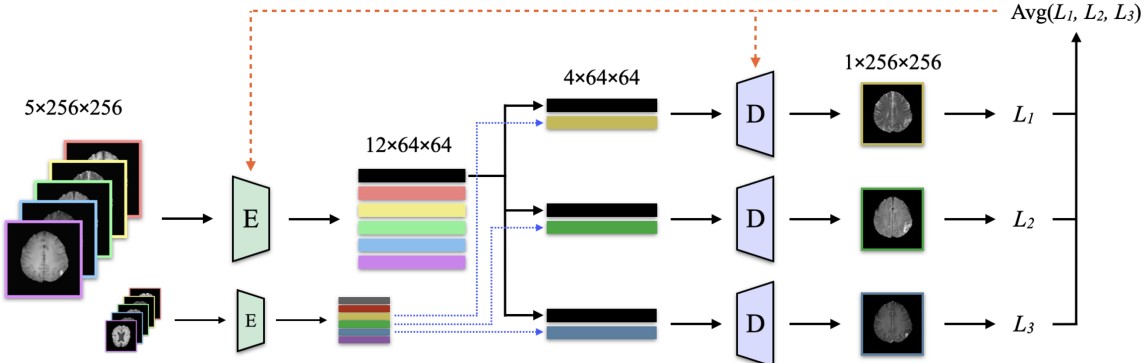

Figure 1: Overall training process of disentangled autoencoder.

## 2. Methods

**Disentangled Autoencoder:** In order to train DLDM capable of generating modality-sharing and modality-specific data, which we call structure vector $z^{struct}$ and style vector $z^{style}$ respectively, an autoencoder that can disentangle those representative features must be trained in prior. We employed four key strategies to achieve this goal. First, if there are $n$ number of MRI modalities, we set the latent dimension $Z$ to have $n+1$ number of representative latent vectors. The additional dimension is for storing the structural information. Second, during the decoding stage, we use pair of $z^{struct}$ and randomly selected single $z^{style}$ for reconstruction. This approach ensures the encoder to separate multi-modal inputs into modality-sharing and modality-specific information. Third, we randomly mix $z^{style}$ with other data inside of a mini-batch. Finally, we average the loss of randomly selected modalities to ensure every modalities having similar reconstruction quality. Figure 1 illustrates the overall training process of disentangled autoencoder.

**Disentangled Latent Diffusion Model:** Our proposed model diffuses and denoises the data in latent space as in latent diffusion model (LDM) (Rombach et al., 2022). The main difference is that DLDM can generate images with feature disentanglement by utilizing aforementioned pre-trained disentangled autoencoder. Specifically, DLDM diffuses and denoises in pair of $z^{struct}$ and $z^{style}$. Also, with this model design, applying class-label $c$ as condition was desirable to selectively obtain the $z^{style}$ of every MRI modality. After training, multi-modal MR images can be synthesized by sending the generated $(z^{struct}, z^{style})$ to the pre-trained decoder, where $z^{struct}$ is fixed and $z^{style}$ varies for different modalities.

## 3. Experiments

**Dataset:** Our proposed model was evaluated using brain metastasis MRI dataset, which is provided by the Department of Radiology and the Research Institute of Radiological Science at Yonsei University College of Medicine. The dataset comprises five sequences, namely T1, T2, FLAIR, WB, and BB. We selected only tumor-containing slices and resized to dimension of 256×256 and normalized from 0 to 1. In total, 13,106 data were obtained in 2D slices, and were then split into 10,484 and 2,622 for train set and validation set,

respectively.

**Evaluation:** To validate that DLDM is capable of generating multi-modal MRIs, the generated samples were evaluated with following metrics: clean-FID and improved precision & recall. Clean-FID is to measure the distance between distributions of real and generated data and improved precision & recall is to measure sample coverage. We compare the results of DLDM with other widely used GAN-based generative model, StyleGAN2 (Karras et al., 2020). The number of generated samples of DLDM and StyleGAN2 were equally set to 1,000 for the fair comparison.

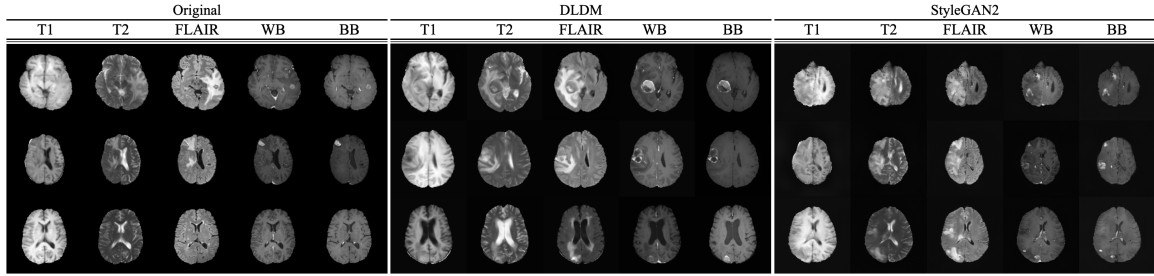

Figure 2: Multi-modal MR image samples generated with DLDM and StyleGAN2. The original MR images are shown on the left for comparison.

**Experimental Results:** The qualitative comparison is presented in Figure 2, which demonstrates that DLDM generates more realistic images than StyleGAN2 in multi-modal MR image generation in human perception. The observation from this comparative study reveals that the generated samples from StyleGAN2 exhibit frequent checkerboard artifacts and unrealistic textures, whereas samples from DLDM closely resemble original MR images, making it hard to distinguish between the two.

For the quantitative results, the average clean-fid over all sequences was 0.00070 and 0.00293 for DLDM and StyleGAN2, respectively. Also, the average precision & recall of DLDM (precision: 0.91208, recall: 0.96021) showed better result than StyleGAN2 (precision: 0.67771, recall: 0.68229). This results demonstrate that DLDM outperforms Style-GAN2 in both image quality and sample coverage.

## 4. Conclusion

In this paper, we presented a novel framework, called DLDM, which leverages the strengths of the diffusion model with feature disentanglement to generate multi-modal brain tumor MRIs. DLDM can generate structure and style vector separately, eliminating the need for a source image when fixed structure is desired. We demonstrated that the samples generated by DLDM exhibit high fidelity and diversity, surpassing the performance of the widely adopted GAN model, StyleGAN2. Thus, we believe that data shortage problem in multi-modal MR images can be solved by using our novel approach.

## Acknowledgments

This research was supported by a grant of the Korea Health Technology RD Project through the Korea Health Industry Development Institute (KHIDI), funded by the Ministry of Health & Welfare, Republic of Korea (grant number: HI21C1161).

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
