# OpenReview forum: "Generation of Multi-modal Brain Tumor MRIs with Disentangled Latent Diffusion Model"
_MIDL.io/2023/Short_Paper_Track — MIDL 2023 Short paper track Poster_

### Official Review · Reviewer_SFGm · 2023-04-21

**Rating:** 6
**Confidence:** 4

**Review:**

This paper proposes multi-modal brain tumor Magnetic Resonance Imaging (MRI) generation framework.
The paper is generally well written with the methods being easy to understand and follow.
However, there are some minor issues needed to be addressed in the revision:
1. It will be better to draw a flowchart for the proposed method.

---

### Official Review · Reviewer_ENWh · 2023-04-24
**Review of: Generation of Multi-modal Brain Tumor MRIs with Disentangled Latent Diffusion Model**

**Rating:** 6
**Confidence:** 3

**Review:**

The authors train a diffusion model on an autoencoder's latent space (similar to latent diffusion, Rombach et al 2022). However, in this latent space the authors have already split the image into modality specific and modality agnostic components. By varying the diffusion parameters for the various components the authors can variably generate in multiple modalities. These are evaluated using an FID variant, and precision and recall.

Overall I am pleased to see latent-space diffusion coupled with style disentanglement used in this context. The results are not surprising, but seem correct from the short description, and I would otherwise be in support of accepting this abstract. However, I have concerns about P&R being applied here: the justification of "measure sample coverage" does not adequately describe the procedure. I am forced to assume that the authors are using the precision and recall of Sajjadi et al. 2018 (citation below), because otherwise it isn't really clear how P&R could be applied.

Sajjadi et al 2018 "Assessing generative models via precision and recall." Advances in neural information processing systems 31 (2018).

I think also it's important to validate the separation of modality specific information; can we swap T1 and T2 modality information but keep non-modality information constant and receive valid images? While visually the images look correct (albeit very very small), this should be assessed numerically.

I feel that this would make good discussion content for MIDL, and thus recommend it for acceptance.